# Comment on Dimou et al. Profile of Membrane Cargo Trafficking Proteins and Transporters Expressed under N Source Derepressing Conditions in *Aspergillus nidulans*. *J. Fungi* 2021, *7*, 560

**DOI:** 10.3390/jof7121037

**Published:** 2021-12-03

**Authors:** Ignacio Bravo-Plaza, Miguel Hernández-González, Miguel Á. Peñalva

**Affiliations:** 1Centro de Investigaciones Biológicas Margarita Salas, CSIC, Ramiro de Maeztu 9, 28049 Madrid, Spain; igbravo@cib.csic.es; 2The Francis Crick Institute, 1 Midland Road, London NW1 1AT, UK; miguel.hernandez-gonzalez@crick.ac.uk

**Keywords:** gene calling, SAR1, ARF GEF, *Ascomycota*, *Aspergilli*

## Abstract

Contrary to the opinion recently offered by Dimou et al., our previously published biochemical, subcellular and genetic data supported our contention that AN11127 corresponds to the *A. nidulans* gene encoding Sec12, which is the guanine nucleotide exchange factor (GEF) specific for SAR1. We add here additional bioinformatics evidence that fully disprove the otherwise negative evidence reported by Dimou et al., highlighting the dangers associated with the lax interpretation of genomic data. On the positive side, we establish guidelines for the identification of this key secretory gene in other species of Ascomycota and Basidiomycota, including species of medical and applied interest.

## 1. Introduction

Sar1 is an essential ARF GTPase of the Ras superfamily acting as a master regulator of COPII vesicle budding from specialized domains of the ER denoted ERES (ER exit sites) [1,2,3]. Like other small GTPases, SAR1 alternates between an inactive, GDP loaded conformation and an active, GTP-loaded conformation. In the latter, SAR1 binds to the ER membrane and recruits effectors from the cytoplasm, most notably the components of the COPII coat, Sec23, Sec24, Sec13 and Sec31. The high affinity binding of GDP to the active site requires the action of a guanine nucleotide exchange factor (GEF), originally baptized Sec12 in the budding yeast [4,5,6], after the Nobel Award-winning screening that led to the identification of the core machinery of the eukaryotic secretory pathway [7]. Indeed, inactivation of the corresponding gene *SEC12* is lethal, as is the ablation/inactivation of the predicted homologues in other eukaryotic cells unless the key role of Sec12 is provided by a paralogue. Sec12 homologues are intrinsically difficult to identify using simple BLAST searches as amino acid sequence conservation is restricted to a relatively small stretch of residues containing the active site.

Until recently *SEC12* had not been characterized in any filamentous ascomycete. On September 2019, we reported that gene model AN11127 from *Aspergillus nidulans* encodes the only Sec12 protein present in this fungus and used its amino acid sequence to infer the identity of the corresponding homologues in other related filamentous ascomycetes [8].

On July 2021, the Journal of Fungi published a paper titled ‘Profile of Membrane Cargo Trafficking Proteins and Transporters Expressed under N Source Derepressing Conditions in *Aspergillus nidulans*’, by Sofia Dimou et al. [9]. On page 11 of this paper these authors discuss their failure to detect Sec12 among the catalogue of secretory proteins analyzed in their proteomic work with the following sentences: 

*“Impressively* (sic erat scriptum)*, all proteins essential for cargo biogenesis were indeed present in the proteome analyzed. The only exception is the product of AN11127, which was recently reported as the Sec12 protein of A. nidulans [52, this refers to our 2019 paper]. However, to our opinion, AN11127 might not be an orthologous protein to the Sec12 protein, which in yeast acts as an essential guanine nucleotide exchange factor (GEF) for activating Sar1p and initiation of COPII vesicle formation. Evidence against AN11127 being an isofunctional Sec12 orthologue comes from the observation that is has no amino acid similarity with Sec12 proteins, and importantly, is not present in several Aspergilli and most ascomycetes.”*

Dimou et al. [9] cast doubts on our report on the identification of the gene encoding Sec12 in the filamentous fungal model *A. nidulans* as the product of AN11127, which might mislead readers to believe that our conclusions were wrong. We shall therefore summarize experimental evidence that overwhelmingly supports our conclusion that the product of AN11127 is indeed Sec12. In addition, we will demonstrate that the statement

*“Evidence against AN11127 being an isofunctional Sec12 orthologue comes from the observation that is has no amino acid similarity with Sec12 proteins, and importantly, is not present in several Aspergilli and most ascomycetes”* is incorrect.

## 2. Results

### 2.1. Compelling Evidence Previously Reported by Bravo-Plaza et al. Demonstrating That the AN11127 Product Has Convincing Sequence Similarity with Other Sec12 Proteins and That It Is Indeed an Isofunctional Sec12 Orthologue

(1) AN11127/Sec12 contains the so-denoted K-loop motif GGGGxxxxGϕxN (where ϕ indicates hydrophobic amino acids) included in an 18-residue conserved sequence (see Figure 1). This motif coordinates a potassium ion crucial for catalysis (i.e., for SAR1 activation) [10,11]. The presence of this motif is highly diagnostic of Sec12 homologues, which is particularly useful when comparing distant Sec12 relatives such as ascomycete and human proteins, as disclosed by Bravo-Plaza et al. [8] in an amino acid sequence alignment included in ref. [8] as ‘S1 Figure’. Note that this alignment includes the founding member of the clan, *Saccharomyces cerevisiae* Sec12p.

(2) An11127/Sec12 contains a PANTHER PTHR23284 ‘prolactin regulatory element-binding domain’, which is the designation for a domain characterizing Sec12-like GEF proteins [8] (http://www.pantherdb.org, accessed on 13 September 2021). This positive PANTHER hit actually includes within it an IPRO15943 WD40/YPTN repeat-like-containing domain (https://www.ebi.ac.uk/interpro/entry/InterPro/IPR015943/, accessed on 13 September 2021). Schlacht and Dacks [12] discussed the issue of low sequence conservation between Sec12 homologues as follows: 


*“This (lack of sequence conservation) is also likely the case for Sec12; low sequence conservation and the presence of multiple WD40 repeats make it difficult to distinguish from other WD40 repeat containing proteins. This became apparent when trying to identify the S. cerevisiae Sec12 using the H. sapiens sequence; multiple rounds of psi-BLAST were required to show that they are indeed homologs, as BLASTp did not provide enough sensitivity to do so”.*


This obstacle possibly contributed to the inability of Dimou et al. [9] to identify sequence similarities between Sec12p and AN11127.

(3) The AN11127 product contains a single, predicted transmembrane helix between residues 458 through 480 [8]. InterPro predicts a protein with an N-terminal cytosolic WD40 domain and a luminal C-terminal region (i.e., a type I protein). This is the expected topology for a Sec12 protein whose active site must reside in the cytosol.

(4) In our 2019 paper [8], we established that AN11127 is an essential gene, as predicted for a Sec12 orthologue.

(5) When AN11127 was endogenously tagged with GFP, such that the fusion protein was expressed under the control of the native AN11127 promoter, we were unable to detect any fluorescence despite the fact that the fusion protein appeared to be fully functional in growth tests, establishing that it was expressed to a level sufficient to permit viability [8] (note that AN11127 is an essential gene). This same fusion protein labelled the ER intensely when expressed under the control of the moderately strong *gpdA^m^* promoter [8], strongly indicating that the AN11127 protein present in very low levels when its synthesis is driven by its own promoter. These low levels expression possibly contributed to the inability of Dimou et al. [9] to detect the protein in their analyses.

(6) *sarA6* is a *ts* mutation in the *A. nidulans* gene encoding SarA^SAR1^ that largely restricts growth and results in reduced protein levels at 37 °C [13]. Forced expression of AN11127/Sec12 partially rescues this growth deficit, indicating functional relationship between SarA/SAR1 and AN11127 and consistent with AN11127 being the GEF for SarA/SAR1 [8].

(7) GFP-tagged AN11127 localizes to the ER, as expected for a SAR1 GEF [8].

(8) The cytosolic domain of the AN11127 protein was expressed in bacteria and subsequently purified. In standard in vitro assays for GEF activity, this protein promoted nucleotide exchange on SAR1, also expressed and purified in bacteria, but not on purified ARF1, a SAR1 relative acting in the Golgi, rather than in the ERES. This specificity is very notable as SAR1 and ARF1 are notoriously similar both in amino acid sequence and structure. Thus, biochemical evidence establishes that AN11127 is a SAR1 GEF [8].

### 2.2. Further Evidence Disproving the Statement by Dimou et al.

*AN11127 has no amino acid similarity with Sec12 proteins, and importantly, is not present in several Aspergilli and most ascomycetes* [9].

(9) FungiDB is a widely used web portal compiling genomic information on Omycetes and Fungi (https://fungidb.org/fungidb/app; accessed on 13 September 2021). The FungiDB search engine automatically detects 113 homologues of AN11127 amongst the Ascomycota. These automatically detected homologues include representatives in each and every order of Ascomycota with the exception of the single-genus order *Pneumocystidomycetales* and the order *Schizosaccharomycetales* (but see below). AN11127-like containing orders include *Mytilinidiales*, *Mycosphaerellales* (*Zymoseptoria tritici*), *Venturiales*, *Chaetothyriales*, *Eurotiales* (22 *Aspergillus* species/strains), *Onygenales* (*Histoplasma capsulatum*, *Coccidioides immitis*) *Erysiphales*, *Helotiales* (*Botrytis cinerea*), *Saccharomycetales* (including Sec12 proteins of *S. cerevisiae*, *Candida albicans* and *Yarrowia lipolytica*), *Glomerales*, *Hypocreales* (*Thrichoderma ressei*), *Magnaporthales* (*Magnaporthe oryzae*), *Microascales*, *Ophiostomatales*, *Sordariales* (*Neurospora crassa*, *Sordaria macroscora*, *Podospora anserina*), as well as in family *Pseudeurotiaceae* (incerte sedis). Thus, the statement by Dimou et al. [9] that AN11127 orthologues are absent in most ascomycetes is entirely wrong (but see also below for additional analysis of *Schizosaccharomyces pombe* disproving this statement).

(10) Automatic searches in FungiDB detected AN11127 orthologues in 19 (*A. nidulans* included) out of 23 genomes of *Aspergilli* present in the database. These 19 sequences corresponded to *A. aculeatus, A. campestris, A. carbonarius, A. clavatus, A. fischeri, A. flavus, A. fumigatus strains A1163 and Af293, A. glaucus, A. lentulus, A. luchuensis, A. niger strains ATCC 1015, ATCC 13496 and N402, A. novofumigatus, A. ochraceoroseus, A. oryzae, A. steynii, A. terreus, A. thermomutatus and A. wentii.* The four ‘Negative genome’ species (i.e., those for which homologues of AN11127 were not detected) were *A. brasilensis*, *A. sydowii*, *A. versicolor* and *A. tubingensis*. One of the two strains and two of the five strains available of *A. luchuensis* and *A. niger*, respectively, were also ‘AN11127-negative’.

(11) We suspected that the inability of Dimou et al. to detect an AN11127 homologue within the proteomes of these seven species/strains of *Aspergillli* was due to the poor annotation/gene calling of some of the fungal genomes. To address this potential shortcoming, we used NCBI’s TBLASTN in FungiDB to search for genomic sequences encoding AN11127 product homologues. Table 1 shows that in each and every instance, homologues were unequivocally detectable in these seven genomic sequences. Therefore, these data totally refute the ‘evidence’ provided by Dimou et al. [9] that AN11127 homologues are absent from ‘several’ *Aspergillus* species, as AN11127 homologues are present in each and every 23 genomes of *Aspergillus* sp. available in FungiDB.

(12) Figure 1 displays a schematic diagram of the domain organization of the AN11127 protein product, depicting the WD40/YVTN repeat-like-containing domain and the location of the GGGGxxxxGϕxN motif. The graph shows amino acid sequence identity across the complete length of the AN11127-like proteins belonging to the species of Ascomycota used for the partial alignment depicted at the bottom. This alignment shows the complete conservation of the GGGGxxxxGϕxN motif. Thus, we addressed the question of whether all the 23 Sec12 protein sequences of *Aspergilli* discussed in (11) contain this diagnostic motif. Notably, the motif was absent from one of the two AN11127 homologues corresponding to the two strains of *A. fumigatus* whose genome is available in FungiDB (gene model Afu5g05910 of strain Af293). However, we noticed that if an alternative gene structure is considered, in which a canonical GT donor site is substituted by a GC dinucleotide, the modified exon-intron structure in the N-terminal region of the gene is such that an ATG located upstream of the previously predicted one restores amino acid sequence identity in the 50 N-terminal amino acids including a perfect match for the active site GGGGxxxxGϕxN motif (Appendix A). Overall, AN11127 and amended Afu5g05910 share a convincing 50.8% identity and 70.3% similarity over a 644 residue overlap (gaps not excluded). These non-canonical GC-AC introns are not uncommon in euascomycetes [14]. Quoting Rep et al. [14], “*These findings* (the existence of non-canonical introns) *have important implications for fungal genome annotation, as the automated annotations of euascomycete genomes incorrectly identified intron boundaries for all of the confirmed and probable GC-AG introns reported here*”. An incorrect exon-intron gene structure in the 5′-UTR/N-terminal region of *A. clavatus* ACLA_009170 in all likelihood explains the absence of the motif in the deduced protein, as high sequence similarity between AN11127 and ACLA_009170 proteins is detectable only after residue 50 of AN11127, reaching 55.1% identity (including gaps in the counting) in the remaining 600 amino acids.

(13) The AFLA_048380 gene product identified by FungiDB as the AN11127 orthologue in *A. flavus* also lacks the diagnostic GGGGxxxxGϕxN motif. Actually, upon close inspection this protein showed to be completely unrelated to AN11127. However, a standard BLASTP search of the *A. flavus* NRRL3357 sequence deposited at NCBI revealed that AFLA_01069 rather than AFLA_048380 is the actual AN11127 orthologue in *A. flavus*, revealing a factual mistake in the annotation and further proving the diagnostic power of this motif.

(14) Our methodology of searching for the presence of the diagnostic GGGGxxxxGϕxN motif plus an IPRO15943 InterPro WD40/YPTN repeat-like-containing domain and a single trans-membrane domain (TMD) was validated with the analysis of the single AN11127 orthologue present in the fission yeast *S. pombe*, the model species of subphylum Taphrinomycotina, which is the most basal subphylum within Ascomycota. The gene is denoted SPO14 (https://www.pombase.org/gene/SPBC3H7.01, accessed on 13 September 2021). A pairwise local align algorithm (https://www.ebi.ac.uk/Tools/psa/lalign/, accessed on 13 September 2021) detects sequence conservation in the region including the completely conserved diagnostic GGGGxxxxGϕxN motif (Figure 1). SPO14 is only 395 amino acid long, with its 325 N-terminal residues matching a WD40/YVTN repeat-like-containing domain and a C-terminal TMD.

## 3. Conclusions

Our previous data, combined with the analysis reported here, indisputably establish that AN11127 encodes a functional Sec12, and demonstrate that functional orthologues are distributed across sequenced genomes of all three subphyla of Ascomycota and in all sequenced genomes of *Aspergilli* deposited in FungiDB. Our analysis also illustrates how a more detailed consideration of bioinformatic data could have prevented Dimou et al. from introducing ‘opinions’ based on negative evidence in their recently published paper in the Journal of Fungi.

## Figures and Tables

**Figure 1 jof-07-01037-f001:**
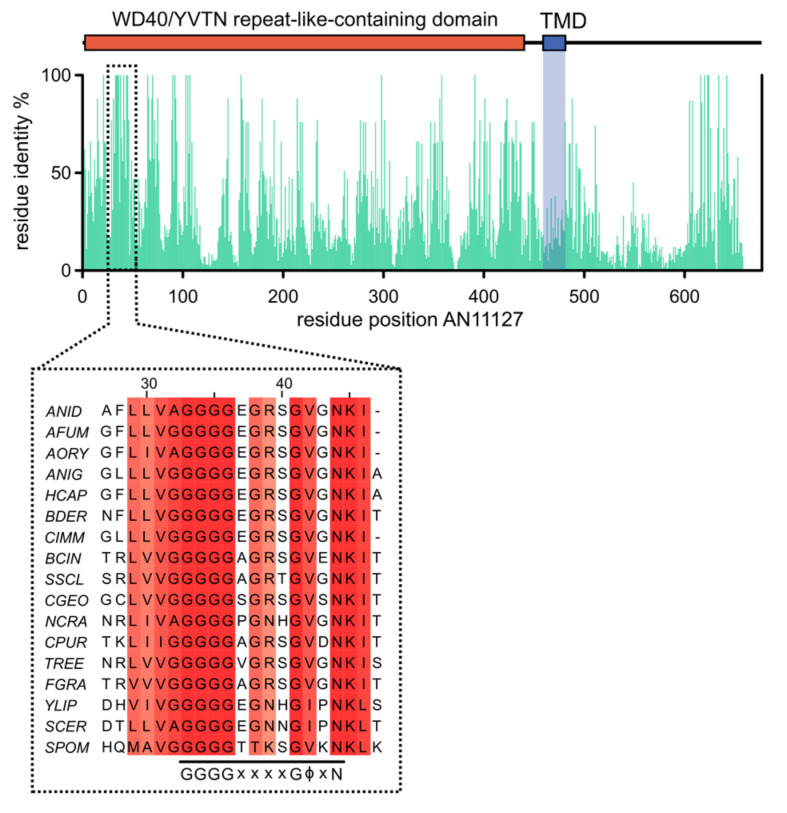
The schematic diagram represents the position of several features over the length of the AN11127 protein sequence: WD40/YVTN repeat-like-containing domain, residues 2-438 (Interpro IPR015943); transmembrane domain (TMD) spanning residues 459-482, as predicted by PolyPhobius. The bar plot represents the percentage of residue identity for each position in the sequence of AN11127, among fungal homologs in *Ascomycota*. Inset: the conserved GGGGxxxxGϕxN K-loop motif (ϕ hydrophobic residue). Multiple sequence alignment with Clustal Omega of AN11127 homologs found in 16 representative *Ascomycota* taxa. Numbers indicate amino acid position for the AN11127 protein sequence. Initials, organism and gene ID: ANID, *Aspergillus nidulans* (AN11127); AFUM, *Aspergillus fumigatus* (Afu5g05910); AORY, *Aspergillus oryzae* (AO090009000671); ANIG, *Aspergillus niger* (ASPNIDRAFT2_1165851); HCAP, *Histoplasma capsulatum* (HCDG_05382); BDER, *Blastomyces dermatitidis* (BDCG_09211); CIMM, *Coccidioides immitis* (CIMG_06226); BCIN, *Botrytis cinerea* (Bcin03g02920); SSCL, *Sclerotinia sclerotiorum* (sscle_03g025460); CGEO, *Cenococcum geophilum* (K441DRAFT_669274); NCRA, *Neurospora crassa* (NCU00379); CPUR, *Claviceps purpurea* (CPUR_05573); TREE, *Trichoderma reesei* (TRIREDRAFT_121774); FGRA, *Fusarium graminearum* (FGRAMPH1_01G22837); YLIP, *Yarrowia lipolytica* (YALI0_A08646g); SCER, *Saccharomyces cerevisiae* (SEC12/YNR026C); SPOM, *Schizosaccharomyces pombe* (spo14/SPBC3H7.01).

**Table 1 jof-07-01037-t001:** AN11127/Sec12 homologues that were not automatically identified by FungiDB.

Sequence ID	Description	Score	E-Value
KV878136	*Aspergillus versicolor* CBS 583.65	856	0
DF126470	*Aspergillus luchuensis* IFO 4308	572	3.0 × 10^−178^
KV878249	*Aspergillus luchuensis* CBS 106.47	565	4.0 × 10^−176^
KV878690	*Aspergillus brasiliensis* CBS 101740	565	6.0 × 10^−176^
ACJE01000012.1	*Aspergillus niger* ATCC 1015	545	4.0 × 10^−169^
KZ851914	*Aspergillus niger* ATCC 13496	545	4.0 × 10^−169^
OGUI01000010.1	*Aspergillus niger* strain N402 (ATCC64974)	545	4.0 × 10^−169^
VTFN01000009	*Aspergillus niger* strain LDM3	543	2.0 × 10^−168^
KV878203	*Aspergillus tubingensis* CBS 134.48	530	8.0 × 10^−164^
An17_A_niger_CBS_513_88	*Aspergillus niger* CBS 513.88	283	3.0 × 10^−79^

## Data Availability

All data are provided in the article.

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
