# Peer review of "Comment on Dimou et al. Profile of Membrane Cargo Trafficking Proteins and Transporters Expressed under N Source Derepressing Conditions in Aspergillus nidulans. J. Fungi 2021, 7, 560"

_jof, 2021, doi:10.3390/jof7121037_

Round 1
Reviewer 1 Report
In the paper comment 'Attention reader: AN11127 does encode A. nidulans Sec12' the authors performed partial bioinformatic analysis convincing that the Aspergillus nidulans AN11127 ORF truly encodes Sec12 orthologue.
The high-throughput proteomic analysis done by Dimou et al. is impressive, they identified of 5690 proteins, which is slightly over a half of predicted coding ORFs (PMID:16372000). It means, that almost half of potentially encoded proteins were undetected. However, due to methodological limitations and probably high cut-off rate, they did not avoid some flaw conclusions. Many proteins are below detection level, since they exist in the cell in a little amount copy number, such as transcription factors, or other, crucial for cell viability. To avoid their toxicity, they are subjected to tight regulation. Since MS/MS analysis did not detect all of the proteins, Dimou et al. went as far as to formulating far-reaching speculations that some undetected proteins does not exist. They also omit a possibility that many of transcripts are spliced non-conventionally by the minor snRNPs U11 and U12 (PMID:8835860).
In my opinion the comparison of AN11127 to S. cerevisiae Sec12, albeit justified, is based on poorly investigated AN11127/Sec12 phylogenetic study.
In S. cerevisiae, where are the two homologous proteins, Sec12p performs a different function than Sed4. Contrary to Sed4p, Sec12p did not bind Sec16p, and genetic tests showed that SEC12 and SED4 were not functionally interchangeable (PMC2199979). That's why at this stage of the study is difficult to precise if AN11127 is closer to Sec12p or Sed4p. Only protein-protein interaction study of AN11127 with orth. Sec16 should discriminates AN11127 function. Despite this, A. nidulans possesses only one gene encoding Sec12/Sed4 homologue, and it is the AN11127 locus. S. cerevisiae Sec12 might forms separate evolutionary branch of Sec12, from that of animal-type or other ascomycetous fungi. However, without thorough phylogenetic analysis, its difficult to undertake semantic speculations if AN11127 is orthologue, paralogue or analogue of Sec12.
There are two things, which might enrich the studies over AN11127/Sec12, missed in PMID:31487505 paper: thorough phylogenetic analysis, and expression data, how the AN11127 gene behaves under altered (e.g. temperature). However, according to RNAseq data from the AspGD database, and other, such as ESTs and TSAs from GeneBank, the AN11127 transcript should be detectable in a splicing form presented by Bravo-Plaza et al.
In the presented comment it is recommended to avoid emotions and focus rather on substantive proofs. The title should be revised, for example to "Evidence supporting that AN11127 encodes A. nidulans Sec12"
point 5. is missed, since the authors did not show low amount of AN11127, but instead localization of GFP-An11127 expressed under strong gpdA promoter.
Author Response
Thank you for the comments and the positive consideration of our note. We have taken two actions:
(i) We have changed the title as suggested and removed from the MS adjectives and expressions that might have been considered 'emotional'.
(ii) point 5: The phrasing of this argument was rather odd. We reworded this point, spelling out in a more reader-friendly phrasing this argument.
We are very grateful for the time that you invested in reviewing our note.
Sincerely,
Miguel A. Peñalva